# De-extinction and poetry

## Katrina Schlunke 🆔

Gender and Cultural Studies, The University of Sydney – Camperdown and Darlington Campus, Camperdown, NSW, Australia

## Perspective

culture; indigenous; poetry; thylacine; history

**Corresponding author:**
Katrina Schlunke;
Email: katrina.schlunke@sydney.edu.au

### Abstract

This article centres a poem concerned with the de-extinction of the thylacine (Tasmanian tiger) to make a wider claim for the importance of poetry as a distinct contribution to thinking about de-extinction. While de-extinction is well understood as a scientific practice, it is also a cultural event. It involves communities with distinct histories who are diversely invested in the idea of extinction, which evoke a range of emotions and embodied responses. A poetry of de-extinction is well placed to situate the science within its complex cultural history while evoking the resistance and multiple temporalities of recorded Indigenous experience. In the instance of the efforts towards the de-extinction of the thylacine (Tasmanian tiger), the colonial acts that led to the original extinction were one part of the violence perpetrated against Indigenous peoples and country.

### Impact statement

This article is concerned with how poetry can extend the frame within which de-extinction is understood. Such an extension is necessary to appreciate how de-extinction is both a cultural and a biological event, with cultural as well as scientific solutions. Poetry can critically enable an affective register and an embodied appreciation of de-extinction, which may offer pathways to alternate ways of understanding extinction more generally. Poetry may also act as a useful form through which silenced perspectives on de-extinction are heard. These may include Indigenous philosophies and the environmental humanities. An expanded, multidisciplinary frame for considering de-extinction offers an enlarged set of possibilities for contribution to the solution to ongoing extinctions generally.

This article begins with a poem written by me in 2024 and reproduced below. The poem is concerned with the de-extinction of the thylacine (Tasmanian tiger), while the following text attempts to make a wider claim for the importance of poetry as a distinct contribution to thinking about de-extinction.

### De-extinction

*On learning that Hollywood actor Leonardo DiCaprio is now supporting Colossal Biosciences along with Marvel and Hunger Games stars, Chris and Liam Hemsworth (and the CIA), to bring back the Tasmanian Tiger/thylacine*

> Something snapped when Leonardo DiCaprio got involved.
> Somehow a man who played Thor was ok.
> This resurrection work was at least including a
> Marvel-ous God.
> But a romantic star – that still innocent face,
> adding his further celebrity gloss to the
> science of so-called return,
> seemed to burn
> the symbolic fuel of forgetting.
> Who else would fear extinction quite so much as
> a creature of the silver screen
> dependent now on streaming services,
> individual exposure, clickbait and
> thirst traps?

The corinna, lagunta, kaparunina (some of the many Palawa words for thylacine) is not extinct.

> It sits in the middle of the creation stories
> from Central Australia – creation dog biting the ear that stills the world.
> And its image in the rock art of Northern Australia still does its work



of organising life.
And in lutrawita nobody forgets.
It is not the corinna that is extinct.

It is the thylacine.
Something named by European naturalists,
transported across the world to scientific papers
then zoos
and then so very quickly Natural History Museums
as 'rare' specimen.
None lived longer than a few years,
many died in transit or upon arrival.
Did the bloke packing up the crates in
Van Diemen's land think; 'transported for life'
as the ship sailed?
Back in England the Royal Society, the Zoological Society and
Very Important Scientific Men
waited to count its teeth and call it primitive
for having a pouch, for being 'non-placental' and so
therefore so
very, very, far from all those
European animals.
Meanwhile back in Tasmania the guns went off,
the traps shut and
island and mainland zoo doors closed
on our very own endlings.

And now the very modern men of not so natural science
are bringing their own invention back to life.
With the hunks of celebrity spunk adding
a fizz of adventure
but also excellent
mediation between fantasy
and an audience.
They are re-creating their thylacine.
A thylacine with its tigerish stripes and doggish form
(so cute, so like Fido the punters will say staring at it in a zoo)
for when it is built where will they release it?

Some wild country where Indigenous care has been
stripped out of the space? Where no Aboriginal rules apply?
The Palawa were always the apex hunters in this so-called
ecosystem
that was wholly Aboriginal country.
That was where the bones of the corinna, like the remains of
kangaroos
were honoured, or incorporated into life
by the building of what Robinson
(in the middle of his own death dealing 'collecting' mission)
called, 'a little house' over
their remains.

It seems there is no plan to enact Indigenous sovereignty
so the corinna can return to country.
There is instead some colonial dream of a mythical
'wild Tassie' where nobody lives.
A *terra nullius* just waiting
for the genetically modified,
manufactured
'specimens'

to turn the land into
a perfect European ecosystem
just how their records
said it should be.

## Thinking with poetry

A particular kind of frustration drove the production of this poem. And a refined version of that frustration shapes this short paratext that sits beside the poem, begging for it to be understood across the disciplinary boundaries of humanities and science, literature and biology, and others that this journal crosses. As a non-Indigenous researcher engaged with Indigenous writings and philosophies to do with Indigenous Country as well as critiques from environmental humanities, which assume humans are not separate from the natural world, the framing of the de-extinction endeavour seemed to be constantly missing that perspective.

While de-extinction is well understood as a scientific practice, it is also a cultural event. It involves communities with distinct histories who are diversely invested in the idea of extinction, which evokes a range of emotions and embodied responses. A poetry of de-extinction is well placed to situate the science within its complex cultural history while evoking the resistance and multiple temporalities of recorded Indigenous experience.

In the instance of the efforts towards the de-extinction of the thylacine (Tasmanian tiger), the colonial acts that led to the original extinction were one part of the violence perpetrated against Indigenous peoples and country. Given Indigenous 'Country' refers not just to land rights but to the interconnections of the more-than-human world, the thylacine is not only in existence in the DNA in laboratories but in continuing Indigenous stories and practices.

Poetry has a particular power to communicate the pathos of the de-extinction experiment. It can call up multiple meanings of extinction itself and gesture towards multiple orders of time. In this way, it can re-imagine what de-extinction might be or mean through a radical re-ordering of assumed knowledges. It can also catch at the collective grief and unspoken hopes that linger within ideas of resurrection and rebirth. Perhaps most powerfully of all, poetry can quite gently show the possibilities of thinking otherwise.

In the instance of the attempted efforts to 'recreate' (Odenbaugh, 2023) the Tasmanian Tiger or thylacine, the questions raised are usually moral or resource-based. That is – 'Should they or should they not?' and/or 'Could those resources be better spent?' The forms available to discuss, and to think about such an important idea, are always imagined as debates – for and against – without a glance at the organising complex that enables the idea in the first place.

As Sara Kianga-Judge, of the Australian Museum, writes:

> A long history of Western separation between what is 'human' and what is 'nature' has left us with a sickness in our relationships with Country. All life depends on Country, so a sickness in those relationships is a sickness that reaches into our health and wellbeing too. This sickness makes us see animals and ecosystems as things instead of beings connected together in life-giving, life-creating, life-supporting patterns. We think this way without even noticing – ask yourself how many times you have referred to an animal as 'it', or in the context of the Museum, how often we haphazardly use words like 'specimens', 'collections', and 'objects' when talking about animal bodies. As a Walbanja Yuin woman, the idea of Country without personhood, aliveness and identity does not make sense. The animal bodies in the Museum are my Ancestors, family, teachers and friends (2022).

Kianga-Judge's critical point, made from her position within a museum of natural history, is particularly telling in relation to de-extinction. Re-creation (de-extinction) projects depend upon accessing materials from the more-than-human world that have usually been stored within collections within natural history museums. In the case of the thylacine, the sequencing of the thylacine genome that is described as 'Step One' of the 'de-extinction' process was 'extracted DNA from the soft tissue of a 108-year-old, alcohol-preserved thylacine pouch young specimen from Museums Victoria, Australia' (Feigin et al., 2018). What different descriptions and approaches would emerge if this 'young specimen' were understood as family, teacher and friend as Kian-Judge suggests? And how would projects of de-extinction be imagined if a key outcome was the healing of the long invented Western divide between 'nature' and 'human' among non-Indigenous peoples?

Multiple Indigenous peoples across the planet have generously shared the complex embodied philosophy that the word 'Country', in the Australian context, evokes, but science, as Moggridge suggests, often fails to re-engage with that 'old knowledge set' (2022). It is here that poetry may be able to fulfil a role as a diplomatic interlocutor able to perform diverse perspectives within orders of affect that may reach otherwise indifferent audiences. Poetry may also help the re-ordering of the frame through which de-extinction projects are understood. For example, Wright commenting on poet Zaina Alsou's use of 'de-extinction' in her poetry clarifies that,'… potentiality for life is recoverable, regardless of place or time, from what injustices have been exacted by colonial logics, the weaponry of empire. The use of the phrase 'de-extinction', and not un-extinction, is a critical distinction: un-extinction would only bring back to life, whereas poetic de-extinction restores life and undoes the conditions of that extinction…' (Wright, 2020). Or as Tabak suggests, 'The poetic not only reflects but also creates' (Tabak, 2024).

Poetry can approach the established scene of de-extinction and ask – not only 'What is de-extinction?' but also 'What is a thylacine?' Palawa kani is the language of the Palawa or Indigenous peoples of Lutrawita/Tasmania, and their word for thylacine is kaparunina. As the Tasmanian Aboriginal Centre (2025) states: 'As a consequence of the devastating impacts of invasion and colonisation on every aspect of our lives, we have had to deliberately and arduously restore our language to its spoken life'. This successful language restoration program connects to the point that Moggridge – an earlier member of the Threatened Species Recovery Hub and Kamilaroi man – that: 'If species have gone, potentially language goes with it, potentially a totem goes with it, potentially a food source goes with it. …these sort of aspects are not considered in species management' (Krebs Lecture, 2022). Moggridge made these comments as part of a panel considering the question, 'Is de-extinction the solution?'. The restoration of language as one part of the ongoing renaissance of the Palawa peoples means that Country is also being revitalised within a recognition of the impact of colonial violence. As Berk writes of Palawa kani, the relative success of the Palawa kani program should be evaluated in relation to the community's history and historicity. While many cultural elements have been revitalised or reactivated, many have been irretrievably lost (2017:19).

Given the dynamic nature of Indigenous Country, Moggridge's later question regarding the reintroduction of resurrected thylacines is very powerful: 'Is Country ready for it?' (Moggridge, 2022). As

Palawa poet *puralia-meenamatta* (Jim Everett) writes in his poem, 'Are You Listening White Australia?' (Everett, 2022).

> …
> for we cannot see your God in Heaven
> nor in his churches who cannot agree
> to whom his pleasure bestows glory.
> For our creator is here with us in this place
> where Moinie is the spirit of All-life
> who connects us like a river's flow
> in a circuitry of ever-cycles with everything
> and the timeless space where we live forever
> within the memory of All-life Country
> with our laws of the land never changing

Perhaps there is a suggestion in this poem that the kaparunina will also 'live forever' within Palawa modes of being and law. And perhaps there is an accompanying ethical question of just how a resurrected thylacine might be fitted within 'the memory of All-life Country' to potentially become kaparunina?

Historically, the kaparunina was also known by other names in different parts of Lutrawita/Tasmania, including coorinna, loarinna, laoonana, and lagunta (Glaskin, 2021, 168). The kaparunina became 'thylacine' through a process of capture, dissection and presentation of description to the European (particularly British) natural history societies, accompanied eventually by 'specimens' sent alive to zoos and dead to museums. The path to becoming a thylacine was often filled with pain and a shortened life. The first kaparunina to be scientifically described was trapped and, 'It remained alive but a few hours, having received some internal hurt in securing it' (Harris 1808). The histories of description and collection are entwined with the histories of bounties and the settler killing of the thylacine. Museums and zoos paid for kaparunina (thus increasing the desire to capture them) and transformed them into thylacine specimens that have eventually become available to scientists to recreate more thylacines. In the particularity of these details is a poetic resource and challenge. How to re-imagine de-extinction while individualising, personalising a particular being? That is how can we evoke feeling and action for the specific while listening well and acting upon Indigenous critiques? And of course, the promise of poetry is that it will do something more than teach. It must also carry an emotional range that is often limited within scientific discourse. Poetry can suggest continuities between the ways that kaprunina were 'invented' as thylacines through the work of natural science networks and the ways new thylacines are being re-invented. And it can suggest a disjuncture between the claims that the habitat of the thylacine 'has remained relatively unchanged' (Pask, 2022), which ignores the pre-colonial care of the country and actions of Indigenous peoples that produced that habitat and are no longer able to do so at the same scale. Suggestion seems a very modest intervention, but perhaps it is the correct scale for the constancy needed to work within poetic de-extinction.

Poetry does not and cannot stand alone as another form that may help us think otherwise about de-extinction and extinction more widely. It is a tiny part of the ecology of thought that requires the widest idea of humanities, creative arts, sciences and technologies to figure responses in these anthropocentric times. Many of those other literary and cultural forms will and have challenged their audience to reflect anew.

Finally, poetry – and the poetry of restoration – may not always appear as expected. Sometimes poetry can take the form of an

Indigenous parable, a story within a story, that makes us reconsider what 'de-extinction' on the ground, on Country, might look like.

In 2019, Ruth Langford, Director of the Nayri Niara Indigenous Festival held on Bruny Island of Lutrawita/Tasmania, recalled a particular experience at a previous festival concerning whales. Whales of all kinds had been slaughtered to near extinction up until the 1890s.

> In the past, Bruny itself has been a real point of trauma due to the great atrocity of whaling. Many felt the whale bones displayed across the island needed to be addressed." Ruth describes how the men made a canoe that year and the women wove a kelp basket. "We loaded the cremated whale bones into the basket and set it into the canoe. Carrying it to the point, we lit the canoe and watched it burn on an incredibly still evening." As a means of explanation she adds, "Aboriginal people tend to cremate their bodies and return elders to dust." Recalling a vivid memory, Ruth finishes, "A stunning umbra came around the sun that evening…and incredibly that next season was the first time a whale mother bought a calf back into the Derwent estuary.

How does 'de-extinction' speak to Country and continuities? How does it acknowledge the colonising science and violent practices it emerges from? Can poetry help convene a different conversation? Perhaps.

**Open peer review.**    To view the open peer review materials for this article, please visit http://doi.org/10.1017/ext.2025.10008.

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
