## [Reviewer Report]

Many thanks to the author for their revisions of the original article. I think the overall argument and the tension between poem-reflection have been greatly improved. With the new introduction to the poem and further reflection on what poetry can do, the article does read as an intervention that balances poetry and academic scholarship very well. There are still some errors in citation across the paper (missing quotations marks, page numbers, etc) and I would recommend a second pair of eyes or copy editor to have a look at the article before publishing.

---

## [Reviewer Report]

De-extinction and Poetry

Does poetry provide us with a distinctive or especially fitting way to think about de-extinction? If so, in what does this special capacity consist? In turn, as an object for reflection and analysis, what is distinctive about de-extinction in general or, more specifically, the project to “bring back” the thylacine, that makes it ripe for poetic exploration? And metaphysically, how does a broadly Indigenous worldview—and the specific worldview ascribed to the Palawa—provide a reframing of the normative questions regarding de-extinction? To my mind, these are the questions raised by the poem and its paratext. I think the questions raised are important enough to warrant publication. What I wish to address here are what seem to me to be the answers suggested by the poem and commentary, as well as what more ought to be addressed.

First, regarding de-extinction itself, it is noteworthy that this piece now explicitly engages with Palawa kani, a language that has been constructed, via a de-extinction process, from the remains of extinct Aboriginal Tasmanian languages (Berk 2017). The poem mentions several terms referring to what Western science renamed the thylacine—corinna, lagunta, kaparunina—and now makes clear that the last of these is the term used in Palawa kani. The revised article draws a compelling connection between species extinction and language extinction, and between their respective restorations. This is a welcome development.

Second, the claim is made that poetry can “gently show the possibilities of thinking otherwise.” To evaluate this claim, we need to go beyond it to the poem itself. Does the poem provide another way to think about extinction and de-extinction beyond what we can glean from the commentary? If so, how does it do it? Arguably, what is distinctive about poetry as opposed to other uses of language is not its propositional content, but *how* it expresses it. In the present poem, sarcasm and wordplay abound, making light of the shallowness of thought undergirding the de-extinction project. Language also plays a performative role. The dubbing of the species as thylacine was a killing act, which ironically leaves the kaparunina untouched, as it lives on in an alternative metaphysics of a different temporal order. What would be brought back would not be the corinna or lagunta, which still exist, but the thylacine, and not even. Overall, I think the poem and commentary make a pretty good case that poetry has something to contribute to thinking about de-extinction in a way that more traditional academic discourse does not, and the revised paratext strengthens this case. One question remains whether this particular poem has the power to do this on its own, or whether it stands in a symbiotic relationship with the paratext.

Third, the revised article now more fully engages with Indigenous philosophy, offering a clearer notion of how to think about extinction and de-extinction and why what might seem to be an act of restorative justice is fundamentally misconceived. For example, in an Indigenous axiology, we must not act autonomously—and capriciously—simply because we can, but rather we must value what it is that the World—and “Country”—call on us to do (Naidoo 2025). There are elements of an Indigenous worldview here, now more fully developed, that prompt further thought and inquiry.

Work cited

Berk, C.D. 2017. Palawa kani and the value of language in Aboriginal Tasmania. Oceania 87(1): 2–20. https://doi.org/10.1002/ocea.5148

Naidoo, R., 2025. Indigenous Australian Philosophy. Philosophy Now 167: 41–45.

---

## [Reviewer Report]

I thank the author for generously engaging my comments and suggestions for revision. I feel the poem itself is able to make a more lasting impression without too much front-loading or contextualisation prior—it is able to affect the reader on its own terms—and the analytical sections which follow are clear and powerful. I appreciate your response to my concerns about the lengthy quotations. I look forward to thinking with this article and poem and I am grateful for the opportunity to have been able to offer modest suggestions for revisions at an earlier stage. For me the article’s standout contribution is not only a justification for the importance of poetry as a political medium within which to imagine technoscience otherwise, but a provocation to those working across the social and natural sciences too, to consider their language carefully and critically. Personally I have taken great inspiration from your provocation to incorporate creative elements in my own writing on de-extinction, species loss, and resistance to technoscientific hegemony.

---

## [Editor Report]

Thanks for your contribution and for your patience. Reviewer 1 mentions minor typographic issues that could indeed be addressed by a copy editor, although it would be great to also see the text gone over carefully prior to submission of a final draft. Reviewers 2 and 3 had no concerns that need to be addressed.